# Extracellular Vesicles May Predict Response to Atezolizumab Plus Bevacizumab in Patients with Advanced Hepatocellular Carcinoma

**DOI:** 10.3390/cancers16213651

**Published:** 2024-10-29

**Authors:** Mara Egerer, Kathrin Schuch, David Schöler, Fabian Artusa, Tobias Püngel, Theresa Maria Holtman, Sven H. Loosen, Münevver Demir, Alexander Wree, Tom Luedde, Frank Tacke, Christoph Roderburg, Raphael Mohr

**Affiliations:** 1Department of Hepatology and Gastroenterology, Campus Virchow Klinikum (CVK) and Campus Charité Mitte (CCM), Charité–Universitätsmedizin Berlin, Augustenburger Platz 1, 13353 Berlin, Germany; mara.egerer@charite.de (M.E.); fabian.artusa@charite.de (F.A.); tobias.puengel@charite.de (T.P.); theresa.holtmann@charite.de (T.M.H.); muenevver.demir@charite.de (M.D.); alexander.wree@charite.de (A.W.); frank.tacke@charite.de (F.T.); 2Department of Gastroenterology, Hepatology and Infectious Diseases, University Hospital Düsseldorf, Moorenstraße 5, 40225 Düsseldorf, Germany; kitti.schuch@gmail.com (K.S.); david.schoeler@med.uni-duesseldorf.de (D.S.); sven.loosen@med.uni-duesseldorf.de (S.H.L.); tom.luedde@med.uni-duesseldorf.de (T.L.); christoph.roderburg@med.uni-duesseldorf.de (C.R.); 3Berlin Institute of Health, 10178 Berlin, Germany

**Keywords:** hepatocellular carcinoma (HCC), immunotherapy, atezolizumab, bevacizumab, prognostic biomarker, extracellular vesicles

## Abstract

This study focuses on extracellular vesicles (EVs) as potential novel biomarkers predicting the outcome of immunotherapy with atezolizumab plus bevacizumab in patients with advanced hepatocellular carcinoma. Firstly, we detected significantly smaller EVs in treatment responders in terms of overall survival. Secondly, a decrease in vesicle size during immunotherapy was related to a longer progression-free survival. Lastly, higher vesicle concentrations and lower zeta potentials were identified as a positive prognostic factor throughout treatment. Our data highlight the potential promising role of EVs as novel biomarkers, potentially helping to identify optimal candidates for checkpoint inhibitor-based treatments for patients with advanced HCC.

## 1. Introduction

Hepatocellular carcinoma (HCC) is the most common form of primary liver cancer and poses a global health challenge, with its rising incidence and limited therapeutic options [1,2]. Recently, the rise of immune checkpoint inhibitors (ICIs) and targeted therapies has transformed the therapeutic landscape of HCC, offering new perspectives, especially for patients in advanced disease stages [3]. The combination of atezolizumab and bevacizumab has demonstrated remarkable efficacy in clinical trials, reshaping the standard of care for advanced HCC [4]. Atezolizumab is an immune checkpoint inhibitor targeting programmed death-ligand 1 (PD-L1), and it reverses T-cell suppression by preventing PD-1 and B7-1 interactions [4,5]. Bevacizumab is a monoclonal antibody targeting vascular endothelial growth factor (VEGF) and inhibiting angiogenesis and tumor growth [4,6]. Anti-VEGF therapies mitigate VEGF-induced immunosuppression in both the tumor and its surrounding microenvironment [7,8] and might amplify anti-PD-L1 efficiency due to the reversion of VEGF-mediated immunosuppression and promotion of T-cell infiltration in tumors [9,10,11]. Notably, bevacizumab itself plays a crucial role in tumor-induced immunosuppression [12]. Nonetheless, despite those therapeutic advances, the heterogeneous nature of HCC and the individual treatment responses observed in patients underscore the urgent need for robust prognostic and predictive biomarkers [1,13]. Identifying individuals who are most likely to benefit from an ICI-based systemic therapy, or who may face treatment resistance, remains a critical clinical challenge [14,15]. More recently, extracellular vesicles (EVs) have gained considerable attention as potential reservoirs of critical biomolecular information that could advance the field of cancer diagnostics and treatment monitoring [16]. Information can be carried by EVs through releasing proteins, lipids, DNA, and RNA into various target cells [17]. These small membrane-bound vesicles, including exosomes and microvesicles, are shed by various cell types into the bloodstream and carry cargo reflecting their cellular origin. Emerging research has demonstrated their pivotal role in intercellular crosstalk, immunomodulation, and the dissemination of tumorigenic signals [18,19,20]. EVs form an integral part of the tumor microenvironment. They are crucial regulators of the immune response during carcinogenesis. They were demonstrated to communicate between fibroblasts, macrophages, and tumor cells. The specific EV characteristics influence the pro- or anti-tumoral role of these molecules [21,22]. Several features of the EVs can be employed for characterization, including size, concentration, morphology, or zeta potential [23,24]. The zeta potential is defined as the surface charge of nanoparticles in solution. Zeta potentials within the range of −10 to +10 mV are regarded as neutral. Nanoparticles exhibiting zeta potentials greater than +30 mV are classified as strongly cationic, whereas zeta potentials below −30 mV are considered strongly anionic [24]. Circulating vesicles might hold the information to unravel the intricacies of HCC’s response to immunotherapy. In this study, we provide evidence that circulating vesicles may hold the key to significantly improving outcome prediction in patients receiving atezolizumab plus bevacizumab—thus opening avenues for optimizing treatment strategies in patients with advanced HCC.

## 2. Methods

### 2.1. Design of Study and Patient Cohort

This retrospective observational study involved 53 patients with advanced, metastatic, or unresectable hepatocellular carcinoma who underwent systemic therapy with atezolizumab and bevacizumab at the Outpatient Clinic of Charité–Universitätsmedizin Berlin between January 2020 and March 2022, as previously reported [25]. The diagnostic criteria for hepatocellular carcinoma (HCC) was based on guidelines from the European Society of Medical Oncology (ESMO) (Lugano, Switzerland), incorporating histological examination, clinical features, or imaging results in patients with liver cirrhosis. [26]. Patients with compensated liver function and an Eastern Cooperative Oncology Group (ECOG) (Philadelphia, PA, USA) performance status of 0 to 2 were deemed eligible for systemic therapy with atezolizumab plus bevacizumab and therefore included in this study. Atezolizumab plus bevacizumab was the systemic first-line therapy for all included patients. A total of 212 longitudinal serum samples for the analysis of EV characteristics were obtained from this cohort. Following the IMbrave150 trial protocol, patients received 1200 mg atezolizumab plus 15 mg bevacizumab per kilogram body weight intravenously every three weeks (Q3W) [4]. Treatment continued until disease progression or occurrence of unacceptable toxicity. In cases of side effects or safety concerns, the treating physician decided on treatment interruption, interruption of individual substances, or complete withdrawal. The study was conducted according to the guidelines of the Declaration of Helsinki and was approved by the local ethics committee of Charité–Universitätsmedizin Berlin (EA2/091/19). The use of an informed consent statement was waived due to the retrospective nature of the study.

### 2.2. Clinical and Laboratory Assessment

All clinical records and lab values were collected at the initial presentation and during the treatment regime every three weeks (Q3W) before the application of atezolizumab and bevacizumab until March 2024. Lab values included Child–Pugh and model for the end-stage liver disease (MELD) scores, alpha-fetoprotein (AFP), C-reactive protein (CRP), neutrophil granulocytes, and lymphocytes and were performed before each therapy cycle. In line with current guidelines, tumor size was assessed by radiological imaging with computer tomography and/or magnetic resonance imaging before treatment induction and every 12 to 14 weeks during the treatment regime [26]. Current response evaluation criteria in solid tumors (RECIST v1.1) were applied to assess response to treatment [27,28]. If a stable response or remission (including complete response or partial response) as defined by RECIST v1.1 was found in the first staging, 12 to 14 weeks after therapy induction, the included patients were considered responders. Patients with progressive disease after 12 to 14 weeks were considered non-responders.

### 2.3. Measurement of EV Characteristics

Blood samples of therapy-naive patients were collected on the day of therapy induction; subsequent samples were collected at regular follow-up appointments (Q3W). Serum was retrieved using centrifugation at 2000 rpm/10 min at room temperature. To minimize the risk of repeated freeze–thaw cycles, serum aliquots were preserved at −80 °C until further analysis. The ZetaView Multi-Parameter Particle Tracking Analyzer (ParticleMetrix, Meerbusch, Germany) was employed to assess vesicle size distribution and concentration. This technique leverages the principles of Brownian motion, widely used for nanoparticle analysis [29,30]. Prior to conducting measurements, the system’s precision was verified by performing an automatic calibration with a nanoparticle standard solution (110 nm diameter) provided by ParticleMetrix. The camera focus was adjusted to ensure that the particles appeared as sharp dots before measurement. The sample anticipated to contain the highest vesicle concentration was used to adjust the camera sensitivity, which remained constant for all subsequent measurements. Samples were diluted in particle-free PBS to achieve a concentration of 1–9 × 10^7^ particles/mL (approximately 200 particles per field of view, PVF). For each sample, three 30 s videos were captured with the script-control function, incorporating a sample flow and a 5 s interval between recordings.

### 2.4. Statistical Analysis

Data are expressed as median with interquartile range (IQR) or frequencies and percentages. The Shapiro–Wilk test was used to test for normal distribution. A two-sample *t*-test was used to calculate significance in cases of normal distribution. Significance for non-normally distributed data was calculated using the Mann–Whitney-U test and chi-squared test. Box-plot diagrams show medians, ranges, and quartiles. Kaplan–Meier analysis was applied to show the effect of vesicle characteristics on overall survival (OS) and progression-free survival (PFS). Differences between responders and non-responders were assessed with the log-rank test. Hazard ratios (HRs) for time-dependent variables were estimated with a stratified Cox proportional hazards model. All probability values were 2-tailed and considered significant when *p* < 0.05. Data analysis was performed using IBM SPSS statistics 29.0 software (SPSS, Armonk, NY, USA). Figures were produced in R (version 4.3.0; R Core Team, Vienna, Austria, 2023) using the package ggplot2 (version 3.4.2; Wickham, UK, 2016).

## 3. Results

### 3.1. Characteristics of Study Cohort

A total of 53 patients receiving atezolizumab plus bevacizumab for advanced HCC were selected for comprehensive EV characterization. A total of 39 patients (74%) showed disease control as defined per RECIST v1.1 in the first staging, 12 to 14 weeks after therapy induction, and therefore were considered treatment responders. A total of 14 patients (26%) were considered non-responders due to progressive disease as defined per RECIST v1.1 12 to 14 weeks after induction. Analyses, including baseline patient characteristics, tumor features, and survival statistics categorized by treatment response status, are shown (Table 1). A total of 47 patients (89%) were male, and the median age at tumor diagnosis was 64 years. A reduced ECOG performance status score of ≥1 was observed in 14 patients (27%). Liver cirrhosis was diagnosed in 43 individuals (81%), while compensated stages characterized by Child–Pugh A classification were detected in more than half of the cohort (*n* = 32, 60%). Among all patients, 23 (43%) suffered from virus-induced, 20 (38%) alcohol-related, and 4 (8%) metabolic dysfunction-associated steatohepatitis (MASH)-induced HCC. A majority (*n* = 36, 68%) was classified as Barcelona Clinic Liver Cancer (BCLC) stage C. Atezolizumab and bevacizumab was the palliative systemic first-line treatment for all included patients, 32% had previous oncologic surgeries, and 53% received locoregional therapies, with the brachytherapy in afterloading technique being the most frequent. Except for macrovascular infiltration (MVI) (*p* = 0.043), neutrophil counts (*p* = 0.026), and neutrophil-to-lymphocyte ratios (NLRs) (*p* = 0.012), baseline characteristics were evenly distributed between responders and non-responders. In terms of clinical outcomes, the overall median survival was 18 months, while the median OS among treatment responders was 19 months compared to 13 months in the non-responder group. The median PFS for the entire cohort was 8 months, with 9 months for responders, and the median duration of response (DOR) was 7.5 months.

### 3.2. Size, Concentration and Zeta Potential of EVs in Patients with Advanced HCC

We analyzed the size, concentration, and zeta potential of EVs on the day of therapy induction, and on follow-up appointments (Q3W). Vesicle characteristics above the 50th percentile were considered large or high, and under the 50th percentile small or low. In therapy-naive patients, the median vesicle size was 204.5 nm (185.4–227.2 nm). The median vesicle concentration was 5.0 × 10^10^ particles/mL (2.2 × 10^10^–1.4 × 10^11^), and the median zeta potential was 63.2 mV (54.7–71.4). We found significant correlations between AFP levels and vesicle size measured 3 weeks (r = 0.310, *p* = 0.04), 6 weeks (r = 0.447, *p* = 0.002), and 9 weeks (r = −0.477, *p* = 0.004) after treatment induction. AFP levels also correlated with vesicle concentration (r = 0.405, *p* = 0.016) and zeta potential (r = −0.416, *p* = 0.013) at week 9 after therapy induction. We subsequently divided the cohort into treatment responders (74%) and non-responders (26%), comparing multiple vesicle characteristics between both groups. In these analyses, no statistically significant differences in size, concentration, and zeta potential were observed at baseline (Table 2). Additional analyses 3, 6, and 9 weeks after therapy induction (Table 2) revealed that vesicle size was significantly smaller among responders (197.0 nm vs. 222.0 nm, *p* = 0.035) at 9 weeks after therapy induction. At this time point (9 weeks after therapy induction), vesicle concentrations tended to be higher among responders, without achieving statistical significance (9.45 × 10^10^ vs. 4.30 × 10^10^ *p* = 0.052).

### 3.3. Decrease in Zeta Potential and Increase in Vesicle Concentration May Predict Treatment Outcome

We next analyzed the longitudinal changes in vesicle characteristics through the first 9 weeks of immunotherapy. We observed an increase in vesicle concentration after therapy induction, whereas vesicle size and zeta potential did not show significant changes after therapy induction. To examine the prognostic and predictive role of longitudinal changes in vesicle size, concentration, and zeta potential, we again subdivided the cohort into responders and non-responders for further analysis. After one applied dose of atezolizumab plus bevacizumab, a decrease in the zeta potential correlated with response to treatment, while non-responders showed a significant increase in the zeta potential (*p* = 0.025) (Figure 1A). Interestingly, no statistical significance was reached for longitudinal changes in the zeta potential when compared with baseline values (prior to immunotherapy induction). For vesicle concentration, similar changes were observed, although they did not reach statistical significance (Figure 1B). In line with these observations, univariate Cox regression analyses revealed that apart from macrovascular tumor invasion (HR: 0.25, 95% CI, 0.0–0.94, *p* = 0.041), only changes in a patient’s zeta potential (HR: 0.15, 95% CI, 0.03–0.88, *p* = 0.035) might serve as a prognostic factor for objective treatment response (Appendix A). No other tumor or vesicle characteristics showed a statistically significant association with objective response to treatment.

### 3.4. Small Vesicles Indicate a Favorable Prognosis

Based on these data, suggesting EV characteristics as an indicator for treatment response, we next tested whether the concentration, size, or zeta potential of EVs predict survival in patients receiving atezolizumab plus bevacizumab for advanced HCC. We first performed Kaplan–Meier curve analyses showing that, in line with our previous findings, a small vesicle size at week 6 after therapy induction indicated a prolonged OS when compared to patients with larger vesicles (*p* = 0.019; Figure 2A). Univariate Cox regression analysis, including various clinicopathological parameters such as tumor stage, markers of inflammation and organ dysfunction, as well as tumor markers, confirmed the role of the vesicle size at 9 weeks after therapy induction as an indicator for OS (HR: 0.18, 95% CI, 0.03–0.93, *p* = 0.040). In addition, an increase in vesicle size within 9 weeks after therapy induction indicated a shorter PFS (*p* = 0.022) (Figure 2B). In contrast, vesicle size, concentration, and zeta potential at baseline could not be linked with the patients’ OS (*p* = 0.567, *p* = 0.767, *p* = 0.343, respectively) or PFS (*p* = 0.753, *p* = 0.965, *p* = 0.317, respectively).

Regarding vesicle concentrations and zeta potential measured at different time points during immunotherapy, no association with OS or PFS could be identified. For example, vesicle concentrations measured 3 weeks (*p* = 0.648), 6 weeks (*p* = 0.804), and 9 weeks (*p* = 0.299) after therapy induction were not linked to patients’ PFS. Zeta potential measured 3 weeks (*p* = 0.188), 6 weeks (*p* = 0.423), and 9 weeks (*p* = 0.080) after therapy induction was not associated with OS. In addition, the deltas of vesicle concentrations and zeta potentials were not indicative of the patients’ prognoses. It is exemplary that neither the delta of vesicle concentration at 9 weeks after therapy induction (Log-rank: 0.124, *p* = 0.725) nor the delta of zeta potential at 9 weeks after therapy induction (9 weeks: Log-rank: 0.018, *p* = 0.893) could predict PFS.

### 3.5. Changes in Vesicle Size Might Serve as a Marker for the Duration of Response

We finally analyzed whether longitudinal changes in EV size might be linked to the duration of response under treatment with atezolizumab plus bevacizumab. Interestingly, a decrease in vesicle size between baseline (therapy-naive patients) and 9 weeks after therapy induction correlated with a longer DOR (HR: 2.62, 95%CI, 0.90–6.90, *p*= 0.051) (Appendix A). Gender (*p* = 0.009) and ALBI-score (*p* = 0.032) were further significant prognostic variables for the DOR. Notably, vesicle size measured 9 weeks after therapy induction correlated with the DOR (r = −0.399, *p* = 0.016). In contrast, vesicle concentrations and zeta potential measured at different time points as well as their longitudinal changes were not associated with the DOR.

## 4. Discussion

The incidence of HCC is increasing, making it a major contributor to the world’s cancer burden [31,32]. Most HCC cases are diagnosed at an advanced and, therefore, unresectable stage where palliative systemic therapy is the only remaining treatment option [4,33]. With the introduction of checkpoint inhibitor-based treatment regimens such as atezolizumab plus bevacizumab or tremelimumab plus durvalumab, the management of HCC has been revolutionized [4,34]. Considering the costs and the potential toxicity of such treatment regimens, it is important to select suitable candidates for these regimens in order to optimize treatment efficacy and diminish adverse effects. Non-invasive and easily accessible biomarkers are urgently needed for personalized oncological decision-making. Nevertheless, current selection criteria predominantly rely on imaging, liver function parameters, and the patient’s general health state, with limited consideration for individual tumor biology and systemic immunological factors [26]. In this context, evaluating the role of EVs prior to and along tumor therapy presents an auspicious approach. EVs have a promising potential as prognostic and predictive biomarkers, complementing conventional assessments of immunological factors such as cell counts and cytokine levels. Especially their modulating role in the tumor microenvironment and their ability to carry immunologically relevant cargo provide a strong biological rationale for their use as biomarkers in the context of immunotherapy [21,22]. Tumor-derived EVs often express immune checkpoint molecules like PD-L1 and thus reflect the tumor’s immunosuppressive status [35]. Monitoring EV characteristics may help predict the tumor’s responsiveness to checkpoint inhibitors. As earlier described, EVs can affect immune cells directly, and also alter the function of antigen-presenting cells [21,22]. A profound understanding of this immune-modulating property of EVs might even pave the way for a novel therapeutic target to modify the efficacy of checkpoint inhibitors. In this study, we aimed to gain insights into the characteristics of EVs in patients with HCC undergoing systemic treatment with atezolizumab plus bevacizumab.

Changes in the concentration and characteristics of EVs have been proposed as novel biomarkers in the context of malignant diseases [36,37,38]. Modifications of vesicle size and concentration were associated with different disease entities such as MASH and alcohol-related liver disease, chronic viral hepatitis B and C infections, primary sclerosing cholangitis, and disease stages like cirrhosis or acute liver failure [39,40,41,42]. For example, alcohol exposure as well as lipotoxicity have been shown to increase the number of circulating EVs, and these EV-loaded biomolecules can further lead to the promotion of inflammation, angiogenesis, and fibrosis in the liver [39]. Furthermore, increased levels of EVs released by T cells and monocytes have been observed in patients with NAFLD [43]. Similarly, our data suggest that changes in EV concentrations and zeta potential are relevant prognostic factors for the treatment outcome among patients receiving atezolizumab and bevacizumab. Responders displayed an increase in vesicle concentration and a decrease in zeta potential. Furthermore, our findings indicate that enlarged circulating vesicles are closely linked to an unfavorable prognosis in patients with HCC undergoing systemic treatment. Patients with smaller EVs exhibited higher OS rates and longer PFS compared to patients with enlarged EVs. This observation of a negative prognostic impact of enlarged circulating vesicles stands in line with the recently published data on enlarged vesicles as a negative prognostic marker in patients receiving transarterial chemoembolization (TACE) for HCC [38].

There is a growing body of evidence underscoring the crucial role of EVs in cancer progression and treatment response [36,38,44,45]. These tiny, membrane-bound structures are known to play a pivotal role in intercellular communication and the transfer of bioactive molecules, making them key players in the tumor microenvironment [46]. The mechanism underlying the association between enlarged circulating vesicles and an unfavorable prognosis in this context warrants further investigation. Enlarged vesicles may be enriched with signaling molecules and immune modulators that potentially influence the effectiveness of atezolizumab and bevacizumab. Additionally, the size of these vesicles may reflect the overall state of immune activation or dysfunction within the tumor microenvironment, providing valuable insights into the host–tumor interaction.

We acknowledge the limitations of our study. Firstly, the number of included patients was relatively low and from a single center. Secondly, we did not include a validation cohort, leading to a lack of statistical power. Thirdly, the presence of immortal time bias cannot be entirely ruled out. Lastly, due to the small cohort size, we were unable to conduct a subgroup analysis and could not determine more prognostic factors than neutrophile count, NLR, and macrovascular infiltration in our cohort. Nonetheless, our findings raise intriguing questions about the potential clinical utility of monitoring EV size as a predictive biomarker in patients with HCC treated with atezolizumab and bevacizumab. EV assessment could serve as a non-invasive tool for patient stratification, identifying those who will likely benefit from this therapy and those who may need alternative treatment approaches, with this finding suggesting confirmation in larger and multicenter trials. In addition to the clinical implications, future research should focus on the underlying mechanisms that link EV size with treatment response. Understanding these pathways may open therapeutic strategies targeting EV-mediated processes.

## 5. Conclusions

In conclusion, our study underscores the significance of enlarged circulating vesicles as a prognostic marker in HCC patients treated with atezolizumab plus bevacizumab. This discovery not only adds to our understanding of the complex interplay between EVs and cancer but also offers a potential avenue for refining patient management and treatment strategies. Further investigations into the underlying mechanisms and validation in larger cohorts are essential to solidify the clinical relevance of EV size as a predictive biomarker in the context of HCC immunotherapy. Ultimately, these insights may contribute to more personalized and effective treatment approaches for HCC patients, improving their overall outcomes and quality of life.

## Figures and Tables

**Figure 1 cancers-16-03651-f001:**
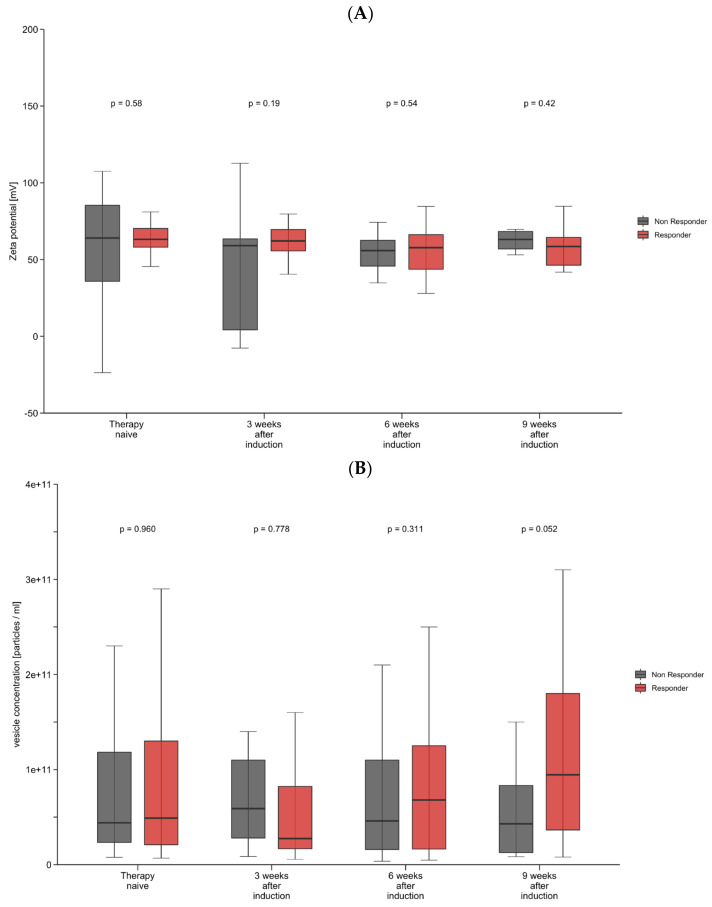
(**A**,**B**) Longitudinal changes in zeta potential and vesicle concentration. Shown are box-plot diagrams of zeta potential (**A**) and vesicle concentration (**B**) at different time points during the first 9 weeks of immunotherapy, comparing responders and non–responders. *p*-Values are from a non–parametric signed–rank test accounting for between- and within–group differences.

**Figure 2 cancers-16-03651-f002:**
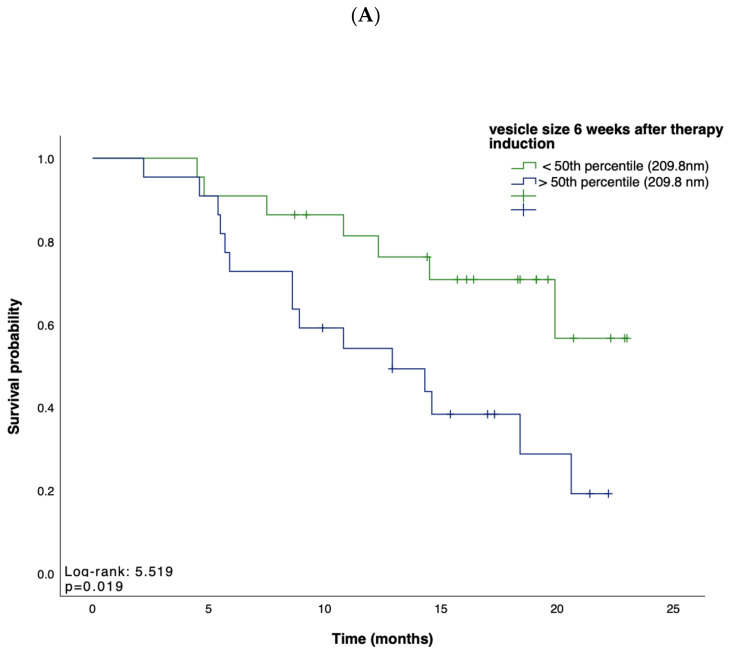
(**A**,**B**) Kaplan–Meier analysis of overall survival and survival without progression. Depicted are Kaplan–Meier estimates of OS according to vesicle size (**A**) and survival without disease progression according to longitudinal changes in vesicle size (**B**). Stratified hazard ratios for death or progression are reported, along with *p*-values. Tick marks indicate censored data.

**Table 1 cancers-16-03651-t001:** Baseline patient characteristics. Depicted are medians with interquartile ranges (IQRs) or counts with frequencies (%). *p*-Values are from two-sample t-test and Mann–Whitney U or chi-squared tests. Non-responders: progressive disease as defined per RECIST v1.1 12 to 14 weeks after therapy induction. Responders: stable disease, or tumor remission. Abbreviations: AFP, alpha-fetoprotein; AL, after loading; ALBI-score, albumin–bilirubin score; ALD, alcohol-related liver disease; BCLC, Barcelona Clinic Liver Cancer; BMI, body mass index; CRP, C-reactive protein; DOR, duration of response; ECOG, Eastern Cooperative Oncology Group; EHS, extrahepatic spread; IQR, interquartile range; MASH, metabolic dysfunction-associated steatohepatitis; MVI, macrovascular invasion; NLR, neutrophil-to-lymphocyte ratio; OS, overall survival; PFS, progression-free survival; RFA, radiofrequency ablation; SIRT, selective internal radiation therapy; TACE, transarterial chemoembolization.

	All Patients (*n* = 53)	Responders (*n* = 39)	Non-Responders (*n* = 14)	*p*-Value
Male *n* (%)	47 (89)	35 (90)	12 (86)	0.683
Age at diagnosis (median/IQR)	64 (58, 71)	63 (58, 70)	68 (60, 73)	0.369
BMI (median/IQR)	26.4 (23.8, 29.6)	26.2 (24.0, 29.0)	28.4 (23.3, 31.7)	0.490
ECOG score *n* (%)				
ECOG 0	39 (74)	29 (74)	10 (71)	0.831
ECOG ≥ 1	14 (27)	10 (26)	4 (28)	0.924
Liver cirrhosis *n* (%)	43 (81)	31 (80)	12 (86)	0.609
Child–Pugh score *n* (%)				
Child–Pugh Score A	32 (60)	23 (59)	9 (64)	0.727
Child–Pugh Score B	11 (21)	8 (21)	3 (21)	0.943
Viral hepatitis *n* (%)	23 (43)	19 (49)	4 (29)	0.129
Hepatitis B	10 (19)	7 (18)	3 (21)	0.775
Hepatitis C	13 (25)	12 (31)	1 (7)	0.078
Hepatitis B/D	1 (2)	1 (3)	0 (0)	0.545
ALD *n* (%)	20 (38)	14 (36)	6 (43)	0.645
MASH *n* (%)	4 (8)	3 (8)	1 (7)	0.947
Barcelona Clinic Liver Cancer Classification *n* (%)				
BCLC B	17 (32)	14 (36)	3 (21)	0.320
BCLC C	36 (68)	24 (62)	11 (79)	0.248
MVI *n* (%)	25 (47)	15 (39)	10 (71)	0.034
EHS *n* (%)	18 (34)	13 (33)	5 (36)	0.872
Previous locoregional therapies *n* (%)	28 (53)	20 (51)	8 (57)	0.709
TACE	16 (30)	12 (31)	4 (29)	0.878
RFA	4 (8)	4 (10)	0 (0)	0.213
AL	21 (40)	14 (36)	7 (50)	0.355
SIRT	2 (4)	2 (5)	0 (0)	0.388
Previous surgery	17 (32)	12 (31)	5 (36)	0.734
Lab values prior to immunotherapy (median/IQR)				
AFP [μg/L]	32.8 (7.8, 561.0)	31.1 (7.7, 658.0)	78.6 (6.8, 742.0)	0.948
Albumin [g/L]	39.5 (34.2, 42.3)	39.1 (34.0, 42.9)	40.7 (35.7, 42.2)	0.521
Bilirubin [mg/dL]	0.8 (0.5, 1.1)	0.7 (0.5, 1.3)	0.9 (0.6, 1.1)	0.348
Lymphocytes [/nL]	1.4 (0.9, 2.3)	1.5 (1.1, 2.2)	1.2 (0.8, 2.0)	0.376
Neutrophiles [/nL]	4.7 (3.1, 6.0)	4.1 (3.0, 5.0)	5.7 (4.2, 7.7)	0.026
NLR score	3.3 (2.0, 4.5)	2.6 (1.7, 4.3)	4.1 (3.1, 6.9)	0.012
ALBI-score	−2.6 (−2.9, −2.1)	−2.6 (−2.8, −2.1)	−2.6 (−2.9, −2.4)	0.791
Clinical outcomes (median/IQR)				
PFS in months	7.6 (3.5, 12.9)	9.3 (7.0, 14.3)	3.2 (2.8, 3.4)	<0.001
OS in months	17.9 (10.6, 23.9)	18.7 (12.8, 22.2)	12.6 (5.6, 28.1)	0.225

**Table 2 cancers-16-03651-t002:** Vesicle size, concentration, and zeta potential in therapy-naive patients and at different time points after immunotherapy induction. Data are shown as medians with interquartile ranges (IQRs). *p*-Values are from Mann–Whitney U test. Abbreviations: IQR, interquartile range.

	All Patients (*n* = 53)	Responders (*n* = 39)	Non-Responders (*n* = 14)	*p*-Value
Vesicle size (median/IQR) [nm]				
Therapy naive	204.5 (185.4, 227.2)	207.9 (184.6, 229.1)	202.6 (184.5, 224.6)	0.626
3 weeks after induction	207.4 (193.1, 216.9)	207.0 (192.0, 223.7)	209.0 (193.5, 213.8)	0.826
6 weeks after induction	209.8 (186.3, 230.3)	211.3 (180.9, 227.2)	203.1 (190.9, 233.9)	0.961
9 weeks after induction	209.9 (187.9, 230.5)	197.2 (186.1, 219.3)	221.8 (208.0, 244.0)	0.035
Vesicle concentration (median/IQR) [particles/mL]				
Therapy naive	5.00 × 10^10^ (2.15 × 10^10^, 1.38 × 10^11^)	5.25 × 10^10^ (2.05 × 10^10^, 1.40 × 10^11^)	4.40 × 10^10^ (2.20 × 10^10^, 1.43 × 10^11^)	0.960
3 weeks after induction	3.00 × 10^10^ (1.68 × 10^10^, 1.10 × 10^11^)	2.80 × 10^10^ (1.65 × 10^10^, 1.04 × 10^11^)	5.90 × 10^10^ (2.00 × 10^10^, 1.20 × 10^11^)	0.778
6 weeks after induction	6.80 × 10^10^ (1.60 × 10^10^, 1.30 × 10^11^)	6.80 × 10^10^ (1.60 × 10^10^, 1.30 × 10^11^)	6.80 × 10^10^ (1.55 × 10^10^, 1.40 × 10^11^)	0.311
9 weeks after induction	5.90 × 10^10^ (1.78 × 10^10^, 1.45 × 10^11^)	9.45 × 10^10^ (2.55 × 10^10^, 1.80 × 10^11^)	4.30 × 10^10^ (1.23 × 10^10^, 9.38 × 10^11^)	0.052
Zeta potential (median/IQR) [mV]				
Therapy naive	63.2 (54.7, 71.4)	63.2 (57.3, 70.5)	64.0 (29.5, 89.6)	0.580
3 weeks after induction	61.3 (44.9, 69.6)	62.4 (55.6, 74.7)	59.1 (4.2, 64.3)	0.191
6 weeks after induction	57.8 (43.7, 70.0)	57.8 (42.6, 67.0)	57.4 (43.0, 73.4)	0.536
9 weeks after induction	60.3 (47.8, 67.3)	58.5 (43.3, 65.1)	63.12 (54.4, 69.2)	0.423

## Data Availability

The data presented in this study are available on request from the corresponding author.

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
