# Peer review of "Extracellular Vesicles May Predict Response to Atezolizumab Plus Bevacizumab in Patients with Advanced Hepatocellular Carcinoma"

_cancers, 2024, doi:10.3390/cancers16213651_

Round 1

Reviewer 1 Report

Comments and Suggestions for Authors

Overall, a well-written paper with an appropriate hypothesis and data analysis. Just some minor suggestions: 1) Figure 1 is blurry and has a random red "A" on it. 2) There are font changes in the first paragraph of the conclusions. 3) Table 2 is difficult to read in terms of the numbers. May be nice to figure out a better way to present the data in this table.

Reviewer 2 Report

Comments and Suggestions for Authors

Dear authors, very interesting manuscript and idea, but  significant limitation of the study are present , which  should be described with more details regarding prognostic factors in both groups. Idea is very interesting but so far from  firm conclusions

Reviewer 3 Report

Comments and Suggestions for Authors

Dear Dr. 

Editor, 

Overall recommendation: 

 Accept with change.

Final comments:

 The authors show extracellular vesicle can predict response to atezolizumab plus bevacizumab in HCC patientsI have some questions.

Major points.

1.     To use extracellular visicles(EVs) as clinical marker for response HCC to atezolizumab plus bevacizumab, ROC curves are necessary. Please provide ROC curve and cut off data of EVs.

2.     Recently, NASH based HCC are not show good response to atezolizumab plus bevacizumab. Are there any correlation between cause of HCC and EVs.

Reviewer 4 Report

Comments and Suggestions for Authors

Dear authors, 

I read with interest your manuscript, which evaluates extracellular vesicles as possible response predictors in patients with advanced hepatocellular carcinoma using atezolizumab + bevacizumab. I have some comments/concerns 

Introduction section

- Authors should better describe the role of extracellular vesicles in tumor immunomodulation and tumorigenic signals since the exposure assessed is related to immunotherapy. Moreover, bevacizumab by the inhibition of VEGF has an immunomodulating action itself. (https://doi.org/10.1016/j.ctrv.2020.102017) 

- Authors should better describe the treatments under study (atezolizumab + bevacizumab), which have two very different biological pathways in the management of cancer patients. Finally, the rationale for combining atezolizumab and bevacizumab is well described in the literature (https://doi.org/10.1016/B978-0-323-90190-1.00008-1) and should be reported to enhance the discussion of the predictors of this combination treatment.

- Please provide more information about zeta potential

Methods

- Authors reported in the methods that 53 patients were included from a previously published study. However, the authors should provide more details about the selection of these patients. Patients could have received a previous treatment before ate+beva intiation?

- Please provide more details about the use COX model. What covariates were included in the model? Multivariate model was not cited in the methods but only in the abstract and results (please report in detail what were the covariates included)

Discussion

- The possibility of incurring an immortal time bias should be discussed in the discussion section. Moreover, looking at survival Kaplan Meiers patients do not seem to die in the first 3 weeks. Is this related to the design of the study (follow-up Q3W), or, for instance, patients dying in the first week after study initiation were considered effectively dead at that precise time?

- More biological rationale should be provided to readers regarding the possible predicting role of EV in the treatment with ATE+BEVA

Round 2

Reviewer 4 Report

Comments and Suggestions for Authors

Dear authors, thank you for addressing all my comments